# Innovative Processes and Technologies for Nutrient Recovery from Wastes: A Comprehensive Review

**Mukhtar Ahmed** [1,2,3,*], **Shakeel Ahmad** [4], **Fayyaz-ul-Hassan** [2], **Ghulam Qadir** [2], **Rifat Hayat** [5], **Farid Asif Shaheen** [6] **and Muhammad Ali Raza** [7]

1 Department of Agricultural Research for Northern Sweden, Swedish University of Agricultural Sciences, 90183 Umeå, Sweden
2 Department of Agronomy, Pir Mehr Ali Shah Arid Agriculture University, Rawalpindi 46300, Pakistan
3 Department of Biological Systems Engineering, Washington State University, Pullman, WA 99164-6120, USA
4 Department of Agronomy, Bahauddin Zakariya University, Multan 60800, Pakistan
5 Department of Soil Science and Soil Water Conservation, Pir Mehr Ali Shah Arid Agriculture University, Rawalpindi 46300, Pakistan
6 Department of Entomology, Pir Mehr Ali Shah Arid Agriculture University, Rawalpindi 46300, Pakistan
7 College of Agronomy, Sichuan Agricultural University, Chengdu 611130, China
* Correspondence: mukhtar.ahmed@slu.se

**Abstract:** Waste management is necessary for environmental and economic sustainability, but it depends upon socioeconomic, political, and environmental factors. More countries are shifting toward recycling as compared to landfilling; thus, different researchers have presented the zero waste concept, considering the importance of sustainability. This review was conducted to provide information about different well established and new/emerging technologies which could be used to recover nutrients from wastes and bring zero waste concepts in practical life. Technologies can be broadly divided into the triangle of nutrient accumulation, extraction, and release. Physicochemical mechanisms, plants, and microorganisms (algae and prokaryotic) could be used to accumulate nutrients. Extraction of nutrient is possible through electrodialysis and crystallization while nutrient release can occur via thermochemical and biochemical treatments. Primary nutrients, i.e., nitrogen, phosphorus, and potassium, are used globally and are non-renewable. Augmented upsurges in prices of inorganic fertilizers and required discharge restrictions on nutrients have stimulated technological developments. Thus, well-proven technologies, such as biochar, composting, vermicomposting, composting with biochar, pyrolysis, and new emerging technologies (forward osmosis and electro-dialysis) have potential to recover nutrients from wastes. Therefore, reviewing the present and imminent potential of these technologies for adaptation of nutrient recycling from wastes is of great importance. Since waste management is a significant concern all over the globe and technologies, e.g., landfill, combustion, incineration, pyrolysis, and gasification, are available to manage generated wastes, they have adverse impacts on society and on the environment. Thus, climate-friendly technologies, such as composting, biodegradation, and anaerobic decomposition, with the generation of non-biodegradable wastes need to be adopted to ensure a sustainable future environment. Furthermore, environmental impacts of technology could be quantified by life cycle assessment (LCA). Therefore, LCA could be used to evaluate the performance of different environmentally-friendly technologies in waste management and in the designing of future policies. LCA, in combination with other approaches, may prove helpful in the development of strategies and policies for the selection of dynamic products and processes.

**Keywords:** waste management; wastes; nutrient accumulation; extraction; and release; biochar; composting

## 1. Introduction

In the era of modern technology and globalization, humankind has reached at its peak in terms of progress in everyday life and activities, such as urbanization, industrial implantation, space technology, and agriculture left the massive amount of solid waste in return. Advancement in lifestyle has facilitated us at best but has taken away the sense of responsibility for our environment, animals, soils, and water bodies. Unfortunately, with every invention, there is no way to go back or cut down its adverse effects on us; for example, heavy machinery usage in agriculture, and the intensive usage of insecticides, pesticides and fungicides. If we only take agriculture as an example, we can easily assess the exploitation of natural resources, poor soil health, and untreated as well as poor quality water with tons of solid waste in the forms of organic, industrial, and sewage sludge. There is a generation of a million tons of soil waste, and management of such wastes is complicated and uneconomical to use at a broad level. Thus, it becomes challenging to keep our surrounding environment clean and healthy. In the United States of America, 6% of municipal solid wastes were recycled [1]. Vietnam produces 27.87 million tons of solid waste annually from different sources, and in its municipal solid waste (MSW) accounts for the largest percentage, i.e., 45.94% [2]. In China, MSW production increases at the level of 3–10% annually [3]. Waste management depends on socioeconomic, political, and environmental factors, thus, it varies from country to country. European countries are shifting toward recycling as compared to landfilling. Different researchers have presented the zero waste concept by considering the importance of sustainability. This concept involves reutilization of organic waste produced from agriculture, municipal, and industrial waste as a resource rather than its disposal as waste directly. Sweden has the best example of resource recovery from waste, and they are using waste to energy technology (incineration) as well as biological treatment to manage municipal waste. Solid waste management is now an environmental as well as a political concern. The potential emerging technology for waste management includes dry composting, sanitary landfill, anaerobic digestion (AD), gasification, pyrolysis thermal processes, plasma arc, bio-chemical conversion, anaerobic process, pyrolysis-gasification, plasma arc-gasification, bioreactor technology, hydrolysis, conversion of solid wastes to protein, and hydro-pulping [4]. Waste management to organic resource utilization and to recover nutrients is key way to have sustainable and eco-friendly agricultural production. Since the food demand of the population is increasing day by day, to reduce the environmental impact of food production, sustainable intensification has been suggested [5]. Nutrient recovery from waste would be a good option to meet the demand of increasing population as fertilizer availability is decreasing day by day.

Nutrients losses in the form of leaching and runoff is worldwide concern but it is difficult to predict the level and fluxes of nutrient losses from the agricultural landscape. The spatio-temporal variability in nutrient losses, weather conditions, farming practices, and water discharge also make it difficult to distinguish between natural causes and agricultural activities. Most of the agricultural production is due to the fertilizer application but half of the applied fertilizer is lost. Nutrients can be lost depending upon their mobility. Mobile nutrients ($NO_3^{1-}$, $SO_4^{2-}$) become unavailable to plants through leaching while immobile nutrients (P, K, and Zn) form chelates with clay/organic material. Low nutrient use efficiency among crops could be due to nutrient loss through leaching (vertical movement) and runoff (lateral movement). Different ways to reduce N losses in Nordic regions were evaluated and researchers concluded that the N application rate and its timing should be in accordance with the need of the crop. Similarly, region-specific solutions and knowledge-based support should also be considered [6]. The main factors controlling phosphorus (P) movement are transport (runoff and erosion potential) and source factors (surface soil P and method, rate, and timing of fertilizer and animal manure applications). Implementation of management practices, such as conservation and contour tillage, cover crops, terracing, buffer and riparian zones, and sediment detention reservoirs can help to control P movement and improve its availability [7]. Excessive nutrient loads is a significant concern, thus, nutrient abatement strategies such as conservation practices and usage of perennial grasses (cellulosic biomass product) should be used. Landscape-management approaches, such

as integration of cellulosic biomass production with agricultural conservation practices (riparian buffers, constructed wetlands, and bioreactors), have shown good potential to reduce nutrients lost through surface runoff [8]. Similarly, Xu et al. (2019) [9] found that growing switchgrass as riparian buffers along cropland can effectively intercept and recycle nutrient runoff from cropland. They also simulated nutrient load reductions through the Soil and Water Assessment Tool (SWAT) model. Nutrient management is the driving force for sustained crop production accompanied by economic sustainability and environmental quality. Both excessive and lesser amounts of application have considerable impacts on the socio-economics of an ecosystem. Decreased quantity may result in reduced crop production failing to feed the projected population of the world, while the incremental nutrient application will pollute the surroundings. Sustainable agricultural production thus necessitates supplementation of those nutrients either through natural processes (nitrogen fixation) or application through animal by-products or mineral fertilizers to crop fields [7–12].

Undesirable wastes are useful tools if managed properly instead of allowing them to contaminate soil, air, and water resources, which create a hazardous environment. The application of technology in this regard can help in agricultural waste management. Untreated animal manures left in the field trigger soil degradation and diminishing air and water quality [13]. Soil amended with untreated municipal solid waste resulted in higher concentrations of Cu, Zn, Cd, and Pb as compared to soil without solid waste application [14]. Nitrogen, phosphorus, and potassium fertilizers are used globally, and are non-renewable. Once applied to the soil, they will be taken up by crop plants exhausting the soil of nutrients and some fraction of it may leach down into the soil. Furthermore, an accelerated increase in prices of fertilizers and required discharge restrictions on nutrients have stimulated the technological development to recover nutrients such as nitrogen, phosphorus, and potassium [15].

Nitrogen (N), phosphorus (P), and potassium (K) are critical nutrients for intensive agricultural production, but their long-term availability and cost of extraction (P and K) is a significant concern for the future. For phosphorus, the primary source is nonrenewable phosphate rock which will be depleted in the future as 90% of phosphate rock reserves are found in just five countries (Morocco, Iraq, China, Algeria, and Syria). Mined P will be exhausted by the end of this century; therefore, it is essential to move to alternative sources, such as organic wastes. N is a renewable resource, but its conversion to ammonia is an energy-intensive and cost dependent process. K availability is also a significant concern as most of the potash ores are in Canada and Europe, which has resulted in its limited distribution globally, particularly for the developing world. Alternative sources of nutrient recovery are required to fulfill the ever-increasing demand of the global population. Since humans and animals consume nutrients from crops and produce nutrient rich waste, this waste from humans can fulfill 22% of the demand of P while animal-derived waste, mainly manure, is widely used as fertilizer. However, the value of nutrient recovery from these wastes is very low, and they also contain heavy metals, pathogenic microorganisms, and odors [15]. Inefficient nutrient management and limited recycling of wastes results in being a major environmental concern. Oxides of N and $CH_4$ are mainly being generated by manure management and excessive use of N fertilizers. It has been reported that 30–32% of GHG emissions were contributed by agricultural activities and livestock production. The other strong concern is eutrophication due to excess nutrients in waterways. Recovery of nutrients from wastes has largely focused on exploiting nutrient cycling reactions and sequestration of nutrients. Recycling nutrients is emerging as an economically sustainable method to solve the issues mentioned above. Nutrient recovery technologies have been used in the past to show the importance of the use of wastes [15,16]. This review focuses on the use of different processes and techniques to recover nutrients from wastes so that the issue of nutrient depletion and climate change can be solved by the use of these technologies.

## 2. Nutrients Recovery Processes (NRPs)

Nutrients recovery processes (NRPs) include the use of composting and vermicomposting as nutrient shortages will be one of the most common problems of the coming years. Since prices of

inorganic fertilizers will increase, therefore, there is a dire need to use waste as a source of nutrient recovery. Furthermore, a mixture of organic and inorganic fertilizer could be an efficient alternative to meet the rising demands of nutrients and solve the issue of food security and climate change. One example of such a mixture is Comlizer (a mixture of ammonium sulfate and composted municipal waste). Recycled human excreta (fertilizer-cum-soil conditioner) and urine (a well-balanced NPK source of nutrients) could be used as a rich source of organic matter and essential nutrients for agricultural crop production. Accumulated domestic wastes are often burnt or disposed of in landfills, which results in the production of a very large quantity of greenhouse gases. This way of waste disposal occupies cultivable land, thus, composting provides a road map to utilize this waste for nutrient recovery complemented with reduced risk of environmental vulnerabilities [17]. Composting is useful in recycling nutrients and managing organic wastes in a sustainable way [18]. Bioconversion of organic matter into a humus-like material called compost is called composting. This process occurs naturally if the required conditions are available. Compost can be used as a soil conditioner as discarded wastes have been decomposed and act as a source of nutrients. Similarly, it can reduce (18%) the amount of wastes entering landfills. A good source of compost could be urban wastes which consist of vegetable matter and dead animals. Roughly, urban areas can generate wastes in the range of 400–800 g per person per day and if the city has a population of five million it could generate wastes greater than 2500 tons per day. Thus, composting could be used as a major portion of these wastes would be vegetables and putrescible matter. Composting can convert a major portion of solid wastes into a marketable product. Compost generated through this process could help to improve the texture of light sandy soil, increase water retention, and enlarge root systems' availability (NPK typical percentage in compost; N, 1.2%; P, 0.7%; K, 1.2%) and uptake of nutrients. Composting is one of the best known recycling processes for the formation of soil conditioner from organic wastes and is the natural rotting or decomposition process of organic matter (crop residues, animal wastes, food garbage, municipal wastes, and suitable industrial wastes) by microorganisms under controlled conditions. It is a good cradle-to-cradle approach as it gives back organic matter to soil taken earlier by plants. This could help to continue the production of healthy crops for sustainable development.. The compost is a rich source of organic matter, and it can play an important role in sustaining soil fertility and crop productivity (Figure 1). The physiochemical and biological properties of the soil could be improved due to the application of compost. Composting types includes aerobic and anaerobic. Aerobic composting gives a more stable organic product which is used dominantly in agricultural production. It was stated that organically-certified compost contained 2–2.5% N [19]. New techniques of waste management have the potential for nutrient recovery if managed properly. The placement of municipal organic waste in landfills will result in emissions of gases causing environmental deterioration. Hence, composting with biochar can be proven better in utilizing the very large amount of waste (15,507–15,888 t day$^{-1}$), including 75% of organic waste as recorded in Bangladesh [20]. The process of composting can be elaborated by the following pathway and chemical equation:

Fresh Organic waste ➡ Aerobic Degradation ➡ Volatilization Anaerobic condition Humus Formation ➡ Compost Stable organic matter

$$Organic\ waste\ (Protein + Cellulose + Lignin) + O_2 \rightarrow CO_2 + H_2O + Compost + Heat$$

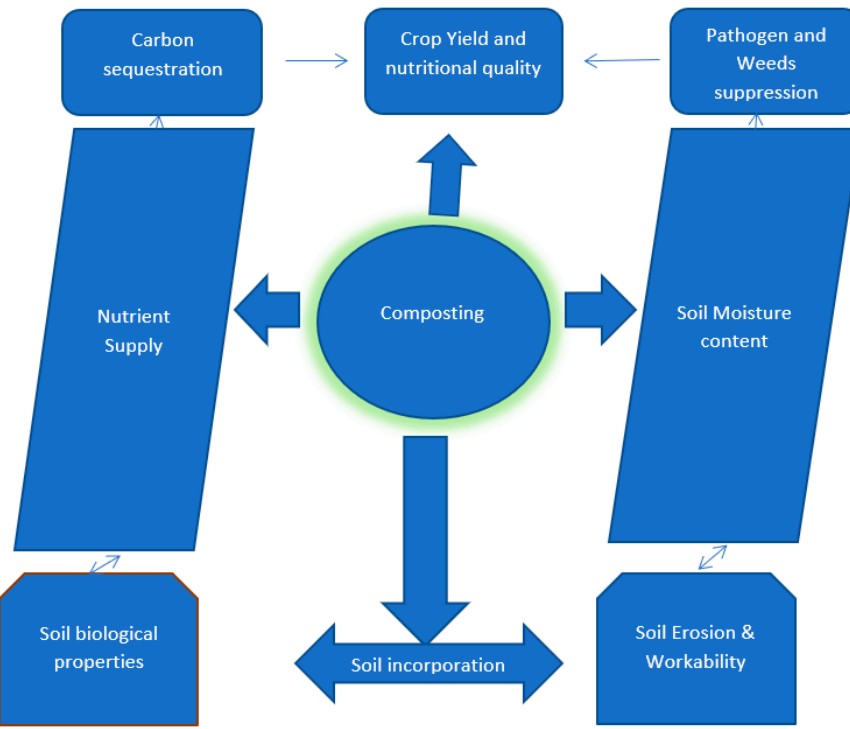

**Figure 1.** Beneficial aspects of composting.

Vermicomposting is a method that utilizes microbes as well as earthworms for the decomposition of solid organic wastes into useful organic manure. It is an ecofriendly and very viable method yet economical to use. It is a bioconversion process, and in this method of treating solid waste, earthworms feed on the organic waste to enhance their population growth rate and synthesis of vermicompost. Vermicomposting can be done for the wastes coming from different sources such as food, plants, animals, pharmaceuticals, and sewage. Its time duration is very important, and usually it takes approximately 28–125 days. As its operational system involves living organisms, for their survival and good quality of vermicompost, several conditions are needed. These conditions are temperature, pH, and moisture content. Generally, it takes a 18–67 °C temperature range, basic pH ranging from 5.9–8.3, and moisture content at 10.68% [21]. Research proves that the process of vermicomposting is a good source of nutrient-rich compost [22–28].

Vermicomposting is a combinable process in which earthworms and other microbes produce useful manures (vermicompost) by reducing waste's harmful effects (Figure 2). The procedure involves earthworms and microorganisms, as mentioned, which shows a mutualistic relationship, and it is better for good biodegradation of waste, maintaining the quality of vermicompost and the nutritional level [29]. The vermicompost main features include higher surface area [30], a low value of carbon to nitrogen ratio, i.e., C:N, maximum nutrients availability, water holding capacity, and increased porosity [31–34].

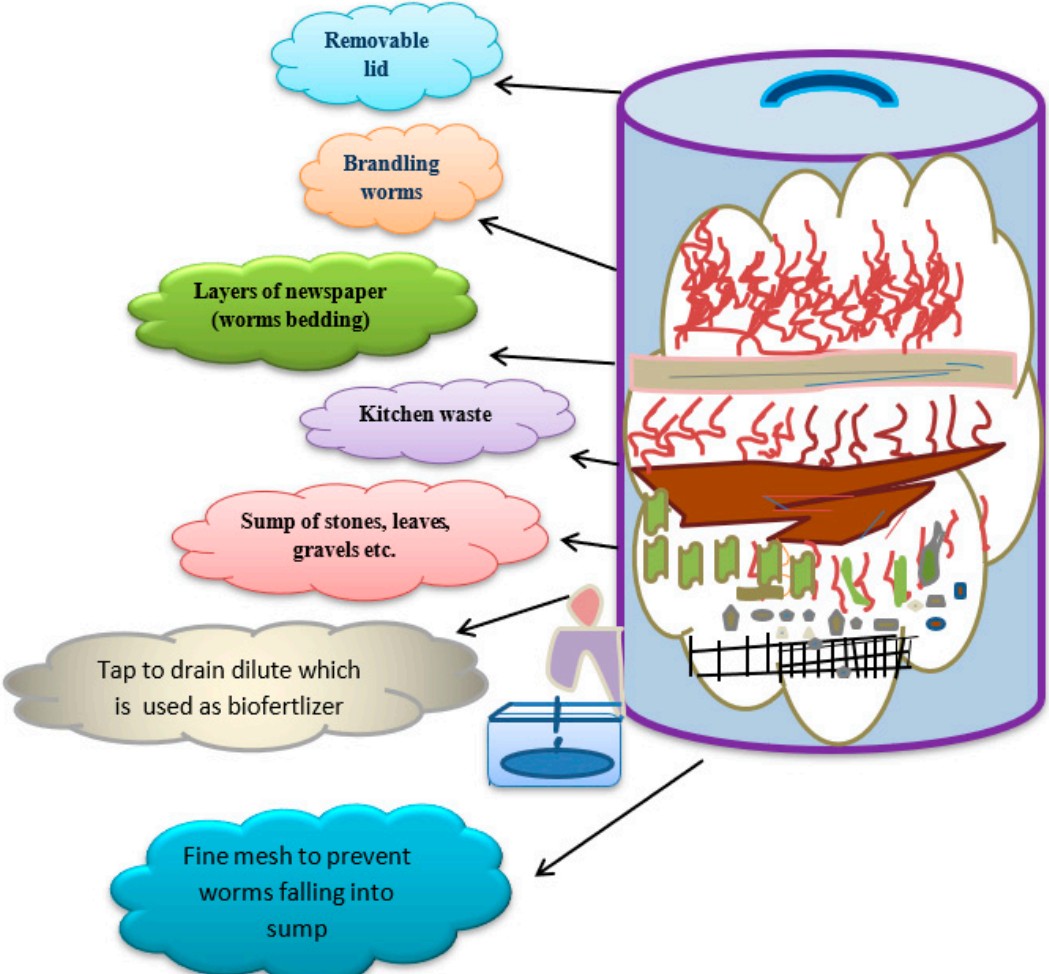

**Figure 2.** Components and procedure of vermicomposting.

Vermicompost fertilizers are enriched with higher concentrations of highly decomposed organic matter, improving soil fertility and crop productivity and can replace synthetic fertilizers. In this way, nutrients can be recycled and detoxified, improving environmental health [35]. The ermicomposting technique comes with total benefit, i.e., its residual matter can be used as bio fertilizer and can be applied on plants, such as maize, cowpeas, soybeans, etc., as vermiwash [36,37].

When animal dung, specifically of cows, gets mixed with the sawdust and guar residues the resulting mixture releases the essential nutrients providing favorable growth conditions for life involved in the biodegrading process [38]. However, sometimes even the addition in waste matter failed to increase the growth, and ultimately the quality, of vermicompost. This manly happens in liquid waste; thus, the efficiency of biodegradation was reduced [39]. Sewage waste mixed with paper waste in a 2:3 ratio increases the efficiency of earthworms and reduces the mortality rate, but the scenario reversed with the addition of pig dung [40]. Good quality vermicompost has been obtained when sewage sludge is used as an initial substrate for earthworms and also biodegraded the aromatic hydrocarbons present in the waste [41]. When apple pomace waste is mixed with straw and used as vermicomposting material and substrate, respectively, it recovers the nutrients from the waste and minerals. It enhances the value of EC from 1.6–4.4 mS/cm, slightly increases the pH from 5.9 to 6.9 and C:N from 13–14 less than 20% (Table 1). This vermicomposting material does not increase the surface area for the activity of earthworms. The resulting vermicompost constitutes N, P, K, and Mg amounts, on average, of about 2.8%, 0.85%, 2.3%, and 0.38%, respectively [42].

**Table 1.** Review of vermicomposting aspects as affected by various factors.

| S. No. | Factors Effecting Vermicomposting | Characteristics of Vermicomposting |
|---|---|---|
| (1) | Degradation rate | Rapid |
| (2) | Temperature | 25–40 °C |
| (3) | PH | Neutral |
| (4) | Humidity | High |
| (5) | Carbon to Nitrogen Ratio (C:N) | <20 °C |
| (6) | Mode of Action | It involves microbes and earthworms. |
| (7) | End product | Stable homogenous fine peat like material and vermiwash which is a liquid portion contains humic acid. |
| (8) | Nutritional status | High due to N, P, K, S and traces of other elements but this mainly depends on what one is feeding to its catalytic agents i.e., worms. |
| (9) | Usage | Not adopted at fullest on industrial level yet. |
| (10) | Capital | High |
| (11) | Shortcomings | It is difficult to maintain its parameter's ranges, such as T, pH, and humidity level. |

Paper cups can be recycled and turned into value-added manure through vermicomposting. This process involves the use of some bacterial groups (*B. endophyticus*, *Acinetobacter baumanni*, *Lactobacillus pantheries, Virigibacillus chiquenigi, Bacillus anthracis, B. funiculus, B. thuringiensis, B. cereus*, and *B. toyonensis*) and earthworm species *Eudrillus eugeinea*, thus, the time duration of biodegradation reduces. Two treatments were used, one is the combination of bacteria with waste paper cups, cow dung, and earthworm *Eudrillus eugeinea,* and the second involves cow dung, waste paper cups and bacterial groups. The resulting vermicompost have C:N value of 15.03% and 11.92%, pH value 8.01% and 7.56% and values of K, Ca, Mg, and P are 1.75% and 1.86%, 50% and 64%, 50.52% and 64.3%, and 46.1% and 51% [43].

A comparative study has been conducted to assess the quality of vermicompost produced by paper waste using an earthworm species *Eisenia fetida* and rice straw. The resulting vermicompost was finer in texture and had high N, P, and K values. There is a low rate of activity reported for earthworms in a treatment of 50% rice straw; the rest all show the significant activity of earthworm. The value of C:N and total organic carbon decrease in vermicompost by 19–102% and 17.38–58.04%, respectively [44]. Better quality of vermicompost is attained when 25% of cattle manure is added to sheep bedding, and the progeny of *Eisenia fetida* grows and develops at a rapid rate [45]. A study was conducted on the quality of vermicompost using liquid waste and tea leaves by utilizing the *Eisenia fetida* species of earthworm. The resulting vermicompost shows declined values for C:N, EC, and total organic matter but the increasing trend has been shown for N, P, and K [46]. The vermicompost made from vegetable waste proved to be nutrient-enriched and very effective. This vermicompost is rich in nutrients like Na, Ca, K, Mg, Fe, Zn, Mn, and Cu [47].

Wine by-product waste can be used as vermicomposting material and provides essential nutrients to the vermicompost which ultimately affect the production of various crops (Table 2). The progeny of earthworms rapidly increases due to this substrate, eventually faster than the rate of biodegradation. The resulting material is like peat and contains polyphenol-enriched extracts in it [48]. As mentioned earlier, kitchen waste is an enriching source of nutrients, and when it is subjected to vermicomposting, the resulting matter recovers the nutrients from it. Amendments in biodegradable waste improve the quality and nutrient content of the resulting product as well as become a good source of feed for earthworms. Thus, wood chips and paper are added into the subjecting vermicompost material. The efficiency of kitchen waste can be enhanced when its pre-composting is done for two weeks at a

temperature of <25 °C. The resulting pre-compost has higher value of nutrients like N, P, and K, i.e., 12, 66, and 40%, respectively, while the end product vermicompost has a range of values of N, P, and K of 2.2–3, 0.4–2.9, and 1.7–2.5 on % dry basis, respectively [28]. The food industry meets the nutritional demands but also produces tons of solid waste which badly pollutes the soil health and water reservoirs. For the management of sewage sludge produced by this industry vermi-technology with *Eisenia fetida* is employed. After three and a half months the resulting vermicompost has higher values of $N_{total}$, $P_{avail}$, and $K_{total}$, i.e., 60–214%, 35–69%, and 43–74%, respectively. The observed C:N value was 61–77% [49]. Likewise, for the food industry, industrial waste was also subjected to vermicomposting for nutrient recovery. In this process *Eisenia fetida* is utilized for good quality vermicompost. A total of nine vermi-reactors were utilized in which different concentrations of industrial waste has been used. In vermi-reactor 9, which had a 100% concentration of waste, earthworms failed to survive. The resulting vermicompost has more heavy metals concentration than before due to mineralization and fragmentation. Low pH and C:N has been reported but an increasing trend was observed for EC, N, P, and K contents. Contents of total Kjeldahl nitrogen, i.e., TKN, has also been increased (12–28 g $kg^{-1}$) [27]. Vermicompost generated from duck manure with reeds, straw, and zeolite as additivess can add up to 236 and 233 $mgg^{-1}$ carbon into the soil. Even after 100 years, carbon potentially remains in the soil with a minimum of 4.72 and 4.66 mg $g^{-1}$ and a maximum with 23.6 and 23.3 mg $g^{-1}$ derived from previously mentioned resources [50].

**Table 2.** Crop production as affected by various vermicomposting treatments.

| S No. | Crops | Treatments | Parameters Affected by the Addition of Vermiwash and Vermicompost | Literature Cited |
|---|---|---|---|---|
| 01 | *Triticum aestivum* | Goat dung and vegetable wastes as additives with the qty. of 10 g $m^{-2}$. | Plants show vigorous growth when vermiwash which is rich in humic acid is applied through foliar spray. Zinc and copper also become available to the plant by the activity of worms and microbes supplied by vermicomposting. | [51] |
| 02 | *Zea mays* | Three levels of vermicomposting, i.e., 0%, 50%, and 100% respectively with same 03 levels of NPK as former. | Crop is more responsive to at 100% of NPK and vermicomposting. This treatment shows maximum height, i.e., 158.22 cm, more leaves per plant, i.e., 11, cob length 17–18 cm, the highest yield of 42.70 $qha^{-1}$, and maximum net return. | [52] |
| 03 | *Cicer arietinum* | Use of vermicompost as fertilizer. | Increased photosynthetic activity reported in gram when subjected to drought. As vermicompost is rich in hormone alike substance humic acid which is known for mitigating the effect of water stress, alleviates the effect of drought on the crop. | [52,53] |
| 04 | *Brassica compestris* | 03 levels of vermicompost, i.e., control, 2.5 and 05 t $ha^{-1}$ have been used along with 05 levels of different nutrients, i.e., Fe, Zn, and S. | Increasing level of vermicompost tends to enhance plant height, no. of siliqua per plant and no. of seeds per siliqua, grain weight, biological and grain yield of this crop. Whereas, the application of mentioned nutrients increases the available nitrogen, phosphorus, potassium, sulfur, zinc, iron, manganese, and copper. It is also influential on the oil content, availability of organic carbon, EC and pH of soil. Combination of both treatments proves to be more beneficial as compared to separate application. | [53] |

**Table 2.** *Cont.*

| S No. | Crops | Treatments | Parameters Affected by the Addition of Vermiwash and Vermicompost | Literature Cited |
|---|---|---|---|---|
| 05 | *Arachis hypogea* | Application of phosphorus enriched vermicompost. | A crop treated with vermicompost that is enriched in P at the rate of 150% with sufficient water conditions resulted in more yield as compared to the treatment utilizing P at 100% with inorganic fertilizer. | [54] |
| 06 | *Oryza sativa* | Priming of seeds with vermicomposting. | Better seed emergence rate and development of early and healthy seedlings. | [55] |
| 07 | *Vigna radiata* | Cow dung with *Eisenia foetida*. | When vermiwash applied at the concentration of 10%, 20%, and 30% it increases the plant growth. It also stimulates the length of hypocotyl and radical. It is responsible for early seedling establishment as well. | [56] |
| 08 | *Vigna mungo* | Vermicompost made up with cattle litter, equine litter, and poultry litter. | Addition of vermicomposting to the soil during the life cycle of this crop resulted in enhanced growth, better combating with water stress, more pods, increase accumulation of protein content, and more biological and grain yield. | [57] |
| 09 | *Helianthus annuus* | Application of vermicompost with inorganic fertilizer. | In water deficit conditions, vermicompost tends to increase the water holding capacity and availability of nutrients lead to improvement of plant growth, more source-to-sink accumulation of assimilates in sunflower. | [58] |
| 10 | *Pennisetum glaucum* | Four levels of vermicompost with RDF, i.e., 60 kg N and 30 kg of P (recommended dose of fertilizer) levels. | Addition of vermicompost with 100% RDF shows a better result than control and 50% RDF. Crop shows more height, number of effective tillers, and grain weight. | [59] |

The vermicomposting method is considered better in comparison to composting in order to kill the pathogens, but some research studies showed that composting that utilizes a high temperature of up to 70 °C has more ability to kill pathogens. As the vermicomposting method includes earthworms, its operating temperature ranges between 30–35 °C, and a temperature above this level may prove lethal to earthworms and may stop the bioconversion process [37] and is, thus, less effective to kill pathogens. A comparison of end product quality between composting vs. vermicomposting were studied by using municipal compost (MC), municipal vermicompost (MV), and backyard vermicompost (BV) (Tables 3 and 4). Microbial biomass-C and dehydrogenase activity was higher in MC while hydrolase activities (urease, protease, and phosphatase) were higher in the vermicomposts than in the municipal compost (Table 4). The result further showed that vermicompost had significantly larger nutrient concentrations than the compost when mixed with the soil, higher microbial population sizes and activity, and increased crop yield [60].

**Table 3.** Composting vs. vermicomposting.

|  | **Composting** | **Vermicomposting** |
|---|---|---|
| Depth | Can be any depth | Worms usually prefer to live in the top 6–12″ of the bedding (cannot be deep) |
| Convenience | Outdoors only with specialized buildings and equipment | outdoors or indoors |
| Speed | Hot composting takes 6–9 months to produce fertilizer | Much faster |
| Heat Levels | Hot as the aerobic breakdown of organic matter releases carbon dioxide and heat, resulting in piles than can top 70 °C | Cooler process with temperatures ranging between 10–32 °C |
| Microbial Populations | Dominated by thermophilic (or "heat-loving") microbes | Dominated by mesophilic microbes |
| Aeration | Turning is required | Turning is not required |
| Cost | Cheap | Needs care for worms protections |
| Financial Value | Cheap | Much greater financial value |

**Table 4.** Biological and biochemical properties of composts and vermicompost [60].

|  | **Municipal Compost (MC)** | **Municipal Vermicompost (MV)** | **Backyard Vermicompost (BV)** |
|---|---|---|---|
| **Microbial biomass C ($\mu$g C g$^{-1}$)** | 1147.00a | 703.00b | 335.00c |
| **Urease ($\mu$mol NH$_4^+$ g$^{-1}$h$^{-1}$)** | 3.54b | 3.90b | 6.11a |
| **BAA-Protease ($\mu$mol NH$_4^+$ g$^{-1}$h$^{-1}$)** | 0.31c | 0.96b | 1.83a |
| **Phosphatse ($\mu$mol PNP g$^{-1}$h$^{-1}$)** | 237.00c | 398.00b | 676.00a |
| **Dehyrogenase ($\mu$g INTF g$^{-1}$)** | 193.00a | 123.00b | 77.00c |

Different letters (a, b and c) in Table 4 within a row indicate significant differences at $p < 0.05$.

## 3. Nutrient Recovery Technologies (NRTs)

The pathway for nutrient recovery from wastes includes three steps, i.e., nutrient accumulation, release, and extraction. Pyrolysis, chemical precipitation, adsorption/ion exchange, algae, liquid–liquid extraction, plants, membrane filtration, and magnetic separation could be used for nutrient accumulation.　However, for nutrient release, biological, thermochemical, and bioleaching processes could be used. Furthermore, for nutrient extraction, chemical precipitation/crystallization, gas permeable membrane, liquid-gas stripping, and electro-dialysis are very effective techniques (Figure 3). Some of the nutrient recovery technologies are discussed below.

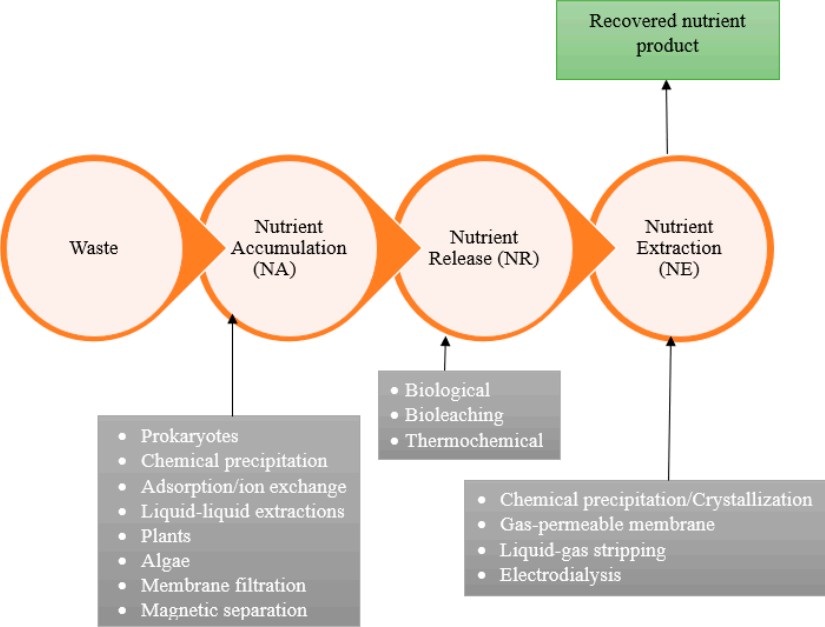

**Figure 3.** Nutrient recovery pathway from wastes.

## 4. Pyrolysis

During the heating process in pyrolysis most of elements are lost to the atmosphere, became soluble oxides, or fixed into a recalcitrant form (Figure 4). For biochar produced from wood under natural conditions the carbon, nitrogen, sulfur, potassium, and phosphorus volatilize around 100 °C, 200 °C, 375 °C, 700 °C, and 800 °C, respectively, while above 1000 °C the volatilization of magnesium, calcium and manganese occur. Biochar produced from sewage sludges at 450 °C contain all P and 50% N [61]. Using organic wastes as such for agricultural practices leads to serious environmental pollution [62]. These wastes can be used as a by-product for the charring process, and the weight and volume of waste also reduces after this process (pyrolysis), which is very important in mostly managing livestock waste [63]. Biochar is a stable form of carbon which can be produced by the controlled heating of animal or plant materials at temperatures of 350–600 °C under a limited supply of oxygen [64]. Almost any materials which are organic can be used to prepare biochar. The benefits of biochar on agriculture and the environment have been presented in Figure 5. Its quality depends upon method of its production and the feedstock used to produce it. Old traditional technologies of producing biochar are energy-consuming and prone to environmental pollution. Biochar can be prepared by pyrolysis, gasification, and hydrothermal carbonization. Pyrolysis (fast and slow) is primary method for making biochar. Slow pyrolysis occurred at a temperature of 400 °C under the absence of oxygen where, as in fast pyrolysis, heating of biomass can be done at 400–700 °C under an anaerobic environment. Modern biochar producing technologies have been developed, including drum pyrolysers, rotary kilns, screw pyrolysers, the flash carbonizer, fast pyrolysis reactors, gasifiers, hydrothermal processing reactors, and wood-gas stoves, all of which produce varying quantities of gas and liquids along with biochar. Biochar application can increase the carbon content in soil and help in carbon sequestration to improve soil and environment quality (Figure 5). Moreover, biochar can also enrich soil with nutrients by improved recycling [65–67]. About more than half of the nitrogen, phosphorus, and potassium can also be recovered [20].

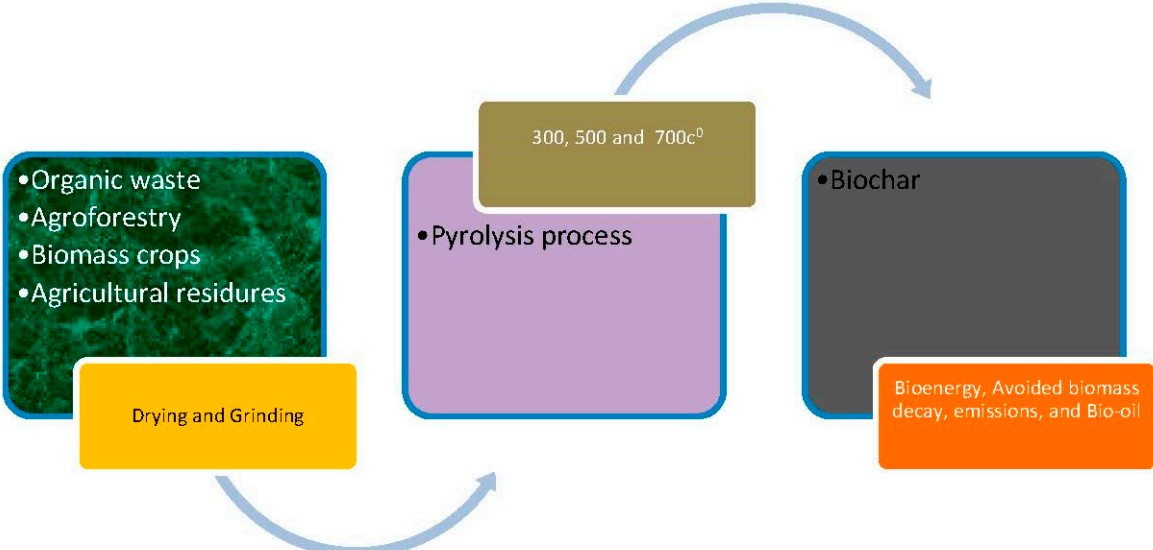

**Figure 4.** Biochar production process.

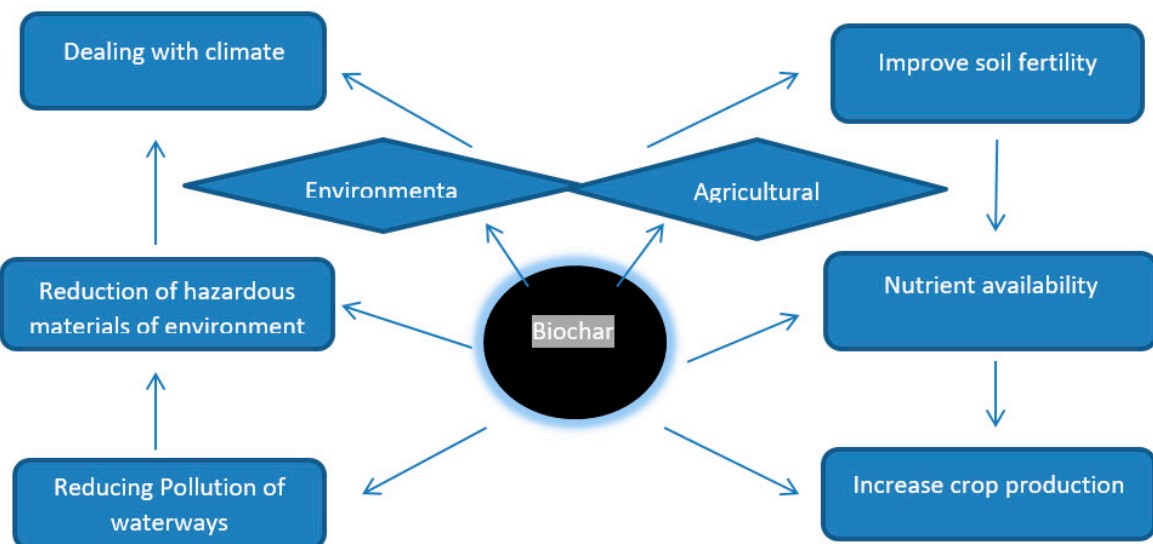

**Figure 5.** Impact of biochar on agriculture and the environment.

Various technologies (electro-dialysis and crystallization) were evaluated for nutrient recycling or recovery based on various aspects as nutrient accumulation (plant microorganisms), extraction (chemical methods), and release of nutrients through bio and thermochemical treatment using waste stream [15]. It was concluded that air and water pollution could be reduced by recovering nutrients without exposure to pathogen risks. Furthermore, the application of these nutrient products in agriculture needs to be developed. Further advancements in innovative technologies for nutrient recovery will help to tackle nutrient losses and combat surge prices of fertilizers needed for sustainability.

The net carbon budget in life cycle of crop can be calculated by the given equation, which includes direct and indirect carbon emissions and organic soil carbon addition:

Net carbon emission:

$$(MgCO_2) = \sum (Ai \times fi)GWP_t + C_{seq}$$

In the above equation, *Ai* represents agricultural inputs; *fi* shows emission factor, GWPt divulges total emissions in two life cycles of maize [68]. The effect of different waste management treatments on nutrient uptake has been presented in Table 5.

**Table 5.** Effect of different waste management treatments on nutrient uptake.

| Treatment | Nitrogen (N) | Phosphorus (P) | Potassium (K) | References |
|---|---|---|---|---|
| Composting of poultry litter with sugarcane and cabbage waste (20–100 days) | N decreased from 26 to 22 g kg$^{-1}$ with increase in composting days | Extractable P decreased with composting time which was higher at early stages | K Increased from 725–775 mg kg$^{-1}$ with increase in composting days | [69] |
| Biochar | No effect | Increased (above ground productivity) | Increased | [70] |
| Only composting | 50% of initial N was found in final compost | 86.4% P was retained at final stage of composting | - | [71] |
| Vermicomposting (plant and animal wastes) | Highest N uptake (168–188 kg ha$^{-1}$) was recorded with 10–20 t ha$^{-1}$ compost application | P uptake was not influenced by direct application. However, rate of 10 tha$^{-1}$ gave highest nutrient uptake (29–37 kg ha$^{-1}$) | Uptake of K was increased | [72] |
| Vermi composting (vegetable waste, mixture of spent mushroom waste, cow dung and leaf litter) | Uptake increased (160 kg ha$^{-1}$) | Increase up to 33 kg ha$^{-1}$ | K uptake (102 kg ha$^{-1}$) decreased as compared to N but was higher than P. | [72] |
| Vermicomposting (mixture of coconut, vegetable waste, leaf litter and cow dung) | Increased up to 168 kg ha$^{-1}$ | Relatively decreased (increased up to 32 kg ha$^{-1}$) as compared to N uptake. | Increased (109 kg ha$^{-1}$) | [72] |
| Vermicomposting (cow dung, leaf litter, vegetable waste and sugarcane) | 142 kg ha$^{-1}$ uptake was recorded | Decreased uptake of P (31 kg ha$^{-1}$) | Decreased (91 kg ha$^{-1}$) | [72] |
| Biochar (rice straw) with nitrogen and phosphorus fertilizers | Increased total uptake by plants up to (166.6 kg ha$^{-1}$) | Increase up to 40 kg ha$^{-1}$ | - | [73] |

## 5. Forward Osmosis

Nutrient recovery from various raw materials, including waste water, can be enhanced through forward osmosis (Figure 6). This is semi-permeable membrane-based technique used for solutions having different concentration, i.e., dilute and concentrated (Figure 6). In forward osmosis, water across the membrane is allowed by osmotic pressure [74]. By using a marine water draw solution it recovered 93% water in the FO process resulting in ten times more recovery of ammonium and phosphates. This high recovery of nutrient is also accompanied by using FO filtration (solution diffusion model) producing 50–80% rejection of ammonium and higher than 90% rejection of phosphate [17]. A higher-strength of nutrient enrichment can be obtained through using a membrane with a high solute selectivity, specifically ammonium and phosphate [74].

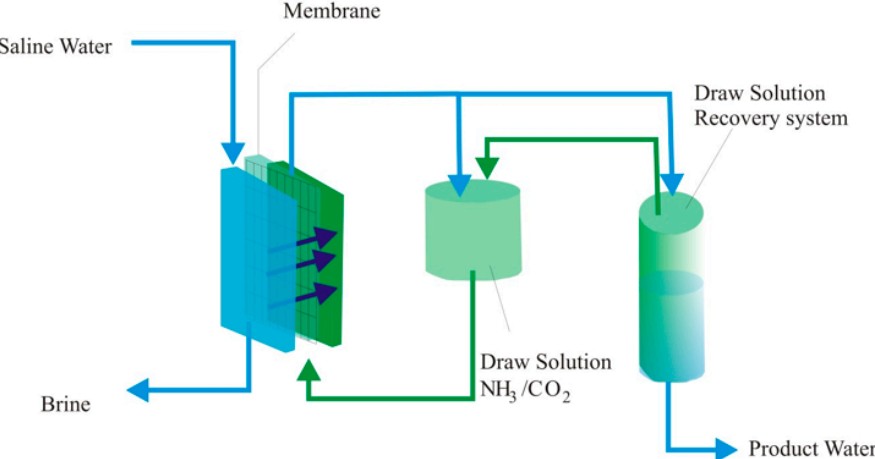

**Figure 6.** Nutrient recovery procedure of forward osmosis.

## 6. Electro-Dialysis

In electro-dialysis (ED), a fraction of nutrients from wastewater to high quality nutrients can be done through arrangement of ion-exchange membranes (Figure 7). Migration of cations and anions towards their respective cathode is driven by direct current field [74]. Applying the ED process using a bipolar membrane to convert phosphate and nitrogen present in sludge to pure phosphoric acid and nitrate or ammonia recovering quantity of (0.075 mol L$^{-1}$) could provide an approach for nutrient recycling [75] with higher recovery efficiency (Figures 8 and 9).

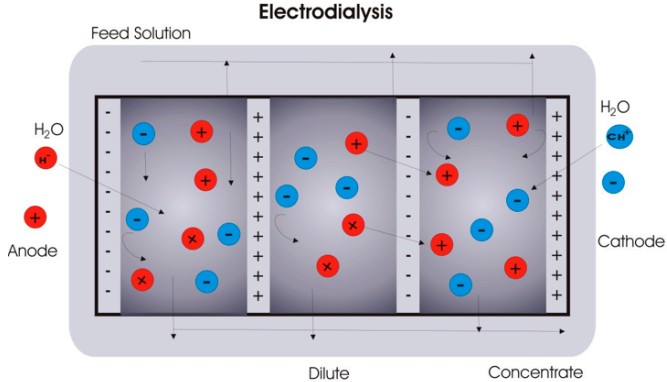

**Figure 7.** Illustration of an electro-dialysis unit mechanism.

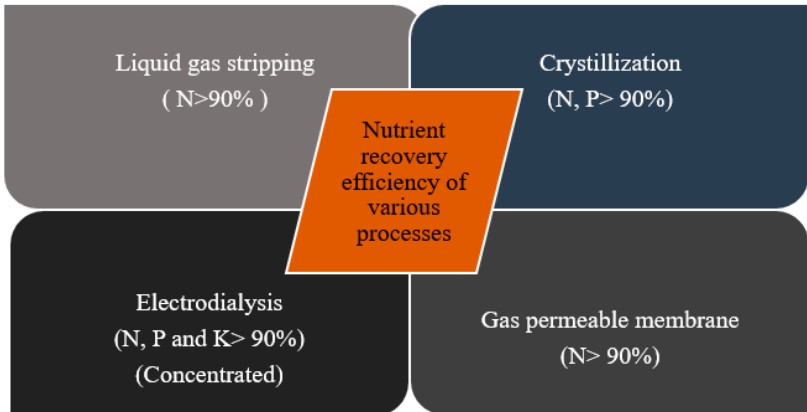

**Figure 8.** Nutrient use efficiency of various techniques.

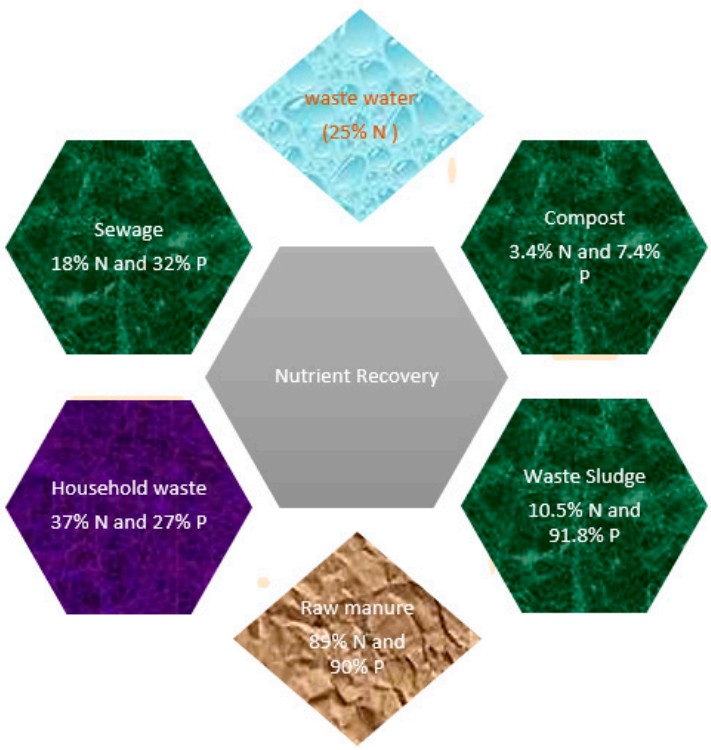

**Figure 9.** Nutrient recovery from various raw materials.

## 7. Quantification of Environmental Impact Using LCA

The impact of product, activity, or process on environment can be evaluated by life cycle assessment (LCA). Nowadays, masses are employing diverse expertise to carry out evaluation of their products for energy gain and environment impacts (Figure 10). Recent LCA studies on various agro-industrial products revealed that agriculture play a key role in the life cycle of various products which can be assisted by the LCA for exploring sustainable development in the future. LCA, in combination with other approaches may prove helpful in development of strategies and policies for selection of dynamic products and processes [76]. Moreover, number of products and by-products either dissipates energy to environment through release of nutrients or is wasted (Figure 10).

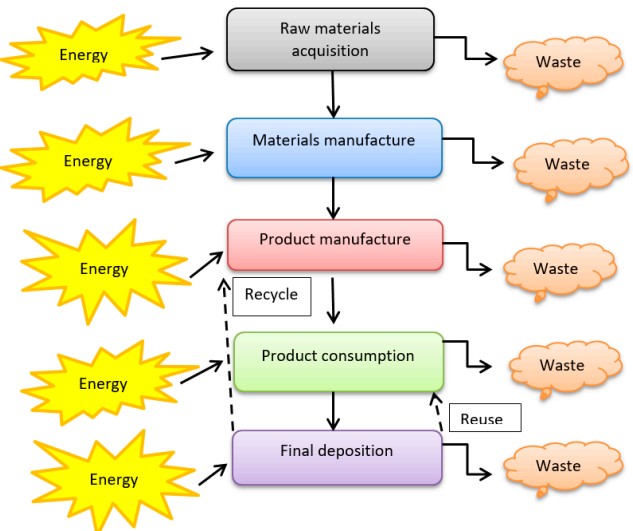

**Figure 10.** LCA Flow chart showing energy utilization and resulting by product.

Biochar contained pyrolysis life cycle assessment was conceded [77] to quantify the magnitude of carbon cycling and profitability of biochar obtained from various agro sources. Regarding emission, reduction was observed with corn fodder showing great economics as compared to forest residue and hence exhibiting the potential for soil carbon sequestration. Greenhouse gas (GHG) emissions observed was reduced due to stable carbon in biochar. This analysis can be used as tool for calculation of biochar environmental pollution and its applications.

Composting was evaluated using life cycle assessment (LCA) for possible impacts on environment. Critical insight to analysis disclosed that compost (processing stage) play crucial role, with largest impact on the environment by emerging emissions to induce eutrophication, acidification and global warming phenomenon. Inference divulged compost as useful for reduction of emissions as compared to peat system. This assessment provides a pathway to explore global impact of emissions on ecosystem and possible minimization co-related with methane, nitrate and nitrous oxides release [78]. The applicability of LCA analysis were reviewed for alternative solid waste treatments practices and results showed that incineration could be suitable for treating wastes. Since it can lead to energy recovery with reduction of GHG emissions [79]. Different waste management options were suggested by researchers and their results indicated that there are several reuse options. This includes the use of the landfill material in a waste-to-energy process after landfill mining, the reuse of the re-gained land in case of landfill mining, the reuse of the capped landfill for energy crop cultivation, and the gasification in a biogas plant in case of a remaining landfill [80].

## 8. Conclusions

Nutrient recovery from wastes is necessary and different techniques, such as pyrolysis, forward osmosis, and electro-dialysis, could be used to recover nutrients. Biochar production processes involve pyrolysis, and it is a very effective method to obtain nutrients from wastes. Biochar can become a more efficient plant growth-enhancing soil amendment as well as a regular animal feed supplement. However, no single technology can effectively recover all nutrients from wastes. Thus, it is necessary to use techniques in combination by considering the nutrient accumulation, release, and extraction/recovery triangle. Economic analysis of the entire recovery process should also be considered as it might be feasible for one location but not for others. Therefore, it is compulsory that nutrient recovery processes must be sustainable with minimum input processes and maximum recovery. Since nutrient management and recovery is interlinked with water and energy issues, energy recovery technologies should be a focus for the future. Similarly, environmentally-friendly technologies, such as composting, biodegradation, anaerobic decomposition, and the usage of biodegradable materials for reducing waste generation should be considered for a sustainable future.

**Author Contributions:** M.A. and S.A. designed the project concept and provided supervision, drafted/wrote the manuscript, collected data. S.A., F.-u.-H., G.Q., R.H., F.A.S., and M.A.R. performed manuscript writing and the collection of data.

**Funding:** This research received no external funding.

**Conflicts of Interest:** The authors declare no conflicts of interest.

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
