# Peer review of "Innovative Processes and Technologies for Nutrient Recovery from Wastes: A Comprehensive Review"

_sustainability, doi:10.3390/su11184938_

Round 1

Reviewer 1 Report

This manuscript provides a nice review of various processes and technologies for nutrients recovery. Overall, this topic is interesting but the writing is sometimes confusing. I just have a few suggestions for the authors to consider prior to possible publication.

Title: please be more specif. Is it “agricultural waste”? If so, refine it with “agricultural waste”

Abstract: similar to title, it is a bit confusing regarding which “waste” this manuscript is talking about. Initially, it starts with that “nutrient management” is important for agricultural systems, then it goes to “treating useless agricultural wastes into useful source”. If so, this manuscripts reviews technologies that can recover N/P/K from agricultural wastes, right? Then it goes to “crude animal manures”. Manure is not a waste, so it is confusing. Suggest authors to re-organize their story to make a more consistent and clear message.

Ln 30: “… technologies for adaption of nutrient recycling” from what? Manure?

Introduction

Overall, the introduction part is lack of focus. This paper focuses on reviewing “technology” for “nutrient recovery from organic wastes”. So focus on this narrative for literature review.

Ln 68 and Ln 92. This paragraph is not necessary. This is a research paper, not a textbook. Please delete it or condense the paragraph into a couple of sentences.

Ln 93-103. What is the relevance of this paragraph to waste nutrient recovery?

Ln 104-113. Irrelevant.

Nutrients Recovery Processes (NRP)

Because this paper discusses nutrient recovery in the context of agricultural production, it is important for the authors to notice that there is a growing interest in “landscape design” approach to reduce nutrient leaching/runoff. For instance, Xu et al. (2018) found that growing switchgrass as riparian buffers along cropland can effectively intercept and recycle nutrients runoff from cropland.

Reference: Xu et al. 2019. Recognizing economic value in multifunctional buffers in the lower Mississippi river basin. DOI: 10.1002/bbb.1930

Quantification of environmental impact using LCA

This portion is outside the scope of this review, and it does not provide useful insights anyway. Suggest authors to delete it, or condense the paragraph into a few sentences as suggestions for future studies.

Author Response

Review Report Form

Open Review

English language and style

( ) Extensive editing of English language and style required 
(x) Moderate English changes required 
( ) English language and style are fine/minor spell check required 
( ) I don't feel qualified to judge about the English language and style 

Is the work a significant contribution to the field?

Is the work well organized and comprehensively described?

Is the work scientifically sound and not misleading?

Are there appropriate and adequate references to related and previous work?

Is the English used correct and readable?

Comments and Suggestions for Authors

Comment#1:

This manuscript provides a nice review of various processes and technologies for nutrients recovery. Overall, this topic is interesting but the writing is sometimes confusing. I just have a few suggestions for the authors to consider prior to possible publication.

Reply to Comments:

Thanks for considering it as nice review of various processes. We have improved the writing to make it clearer and focus to the topic as directed by the reviewers.

Comment#2:

Title: please be more specif. Is it “agricultural waste”? If so, refine it with “agricultural waste”

Reply to Comment:

No, its not only agricultural waste its about all wastes.

Comment#3:

Abstract: similar to title, it is a bit confusing regarding which “waste” this manuscript is talking about. Initially, it starts with that “nutrient management” is important for agricultural systems, then it goes to “treating useless agricultural wastes into useful source”. If so, this manuscripts reviews technologies that can recover N/P/K from agricultural wastes, right? Then it goes to “crude animal manures”. Manure is not a waste, so it is confusing. Suggest authors to re-organize their story to make a more consistent and clear message.

Reply to Comment:

Revised as suggested.

Comment#4:

Ln 30: “… technologies for adaption of nutrient recycling” from what? Manure?

Reply to Comment:

Its from wastes added now.

Comment#5:

Introduction

Overall, the introduction part is lack of focus. This paper focuses on reviewing “technology” for “nutrient recovery from organic wastes”. So focus on this narrative for literature review.

Reply to Comment:

Revised in the lights of comments.

Comment# 6:

Ln 68 and Ln 92. This paragraph is not necessary. This is a research paper, not a textbook. Please delete it or condense the paragraph into a couple of sentences.

Reply to Comments:

This paragraph starting from “Nutrient is any substance which is required for nourishment…………… and carries out sugar translocation has been deleted as suggested by reviewer.

Comment# 7:

Ln 93-103. What is the relevance of this paragraph to waste nutrient recovery?

Reply to Comments:

Deleted as suggested.

Comment# 8:

Ln 104-113. Irrelevant.

Reply to Comments:

Deleted as suggested.

Comment# 9:

Nutrients Recovery Processes (NRP)

Because this paper discusses nutrient recovery in the context of agricultural production, it is important for the authors to notice that there is a growing interest in “landscape design” approach to reduce nutrient leaching/runoff. For instance, Xu et al. (2018) found that growing switchgrass as riparian buffers along cropland can effectively intercept and recycle nutrients runoff from cropland.

Reference: Xu et al. 2019. Recognizing economic value in multifunctional buffers in the lower Mississippi river basin. DOI: 10.1002/bbb.1930

Reply to Comments:

Suggested reference has been added as suggested by the reviewer.

Comment# 10:

Quantification of environmental impact using LCA

This portion is outside the scope of this review, and it does not provide useful insights anyway. Suggest authors to delete it, or condense the paragraph into a few sentences as suggestions for future studies.

Reply to Comments:

We disagree with this reviewer comment as the impact of product, activity or process on environment can be evaluated by life cycle assessment (LCA) and since our aim is to recover nutrients from wastes through different techniques so its good to have LCA analysis of techniques.

Reviewer 2 Report

Sustainability-558992 Review

Title:

Given the lengthy discussion and numerous details and citations of vermicomposting (~ 6/17 pages), the authors should reconsider changing the title to include  ”… review of vermicomposting…”

Abstract

Long, with several misstatements or unclear sentences such as in lines 23-25”.  Nitrogen and potassium and either renewable or inexhaustible nutrients; whereas, Phosphorous is  non-renewable according to the  latest estimates of mine deposits.

The concluding sentence makes no sense, lines 30-33.

Body of the Manuscript

In general, very long, specially the introduction of nearly 3 single-spaced pages with general knowledge information about nutrients such as their well-known chemical forms and physiological functions.

Introduction

Lines 114-115.  Nitrate leaching does not cause soil acidification, but the conversion of ammonia to nitrate does. Excessive leaching can cause soil acidification over time with the loss of base cations, clay minerals….

NRP section

Composting a very common method of green and manure waste recycling, yet the authors do not give it sufficient consideration in light of the amount of space and information that they give to vermicomposting.

In other words,  what are the major advantages and disadvantages of both methods: composting and vermicomposting.

Again, the authors dedicate many pages to describing in detail vermicomposting but only about ½ page on composting with a handful of citations compared to ~20 citations on vermicomposting.  Therefore, the reader is not exposed to a comparative review of these two nutrient recovery processes.

NRT section

Confusing with information on biochar production together with plant nutrient productivity and GH emissions formulas and calculations (not sure what is the aim of this section)

There is also a table comparing management strategies and nutrient uptakes.  But it is far from complete since only 8 citations are in the table that covers composting, vermicomposting and biochar and only cover a very limited number of wastes.

The sections on forward osmosis and electrodialysis seemed misplaced and are more appropriate to an environmental engineering paper.

These two techniques may be innovative (based on old electrochemical principles) but far from being practical when applied to nutrient recoveries from solid mixed organic wastes and have yet to progress beyond laboratory studies, unlike composting and more recently biochar production techniques. These techniques limited to liquids and thus best suited for concentrated industrial waste streams and far from being optimal for complex mostly organic municipal wastewaters. 

LCA section and conclusions

Not sure what this section brings to the manuscript and there are not clear distinctions made since all processes, have carbon emissions footprint.   The authors fail to compare composting techniques with biochar. 

Which processes have the lowest initial, mid and long-term carbon foot-print?

Which techniques limit carbon emissions best and which capture carbon multigene rationally? And finally, which techniques and most cost-effective? (which drives their large-scale implementation beyond lab and field studies).

Conclusions, no section identified

Lines 481 to 484 conclude the manuscript with two vague statements about waste management, threats to the environment and economic analysis.  

Nothing is stated about which of the “innovative” processes mentioned in the manuscript is(are) “best or promising or cost-effective….” to limit environmental damage, nutrient recovery, carbon footprint….etc.….

Recommendations

Authors should focus on solid waste methods of nutrient recycling… primarily composting, vermicomposting, municipal sludges, and biochar. Suggestion: add biosolids too, since these are produced in significant amounts by municipalities and an excellent source of N,P, and micronutrients.

They should endeavor to provide a detailed comparison of these systems of nutrient recovery which should include:

Applicability to types of wastes, complexity, recovery/loss rates (of nutrients), carbon footprint (short and long-terms benefits to carbon storage, life cycle impact, costs, scalability.

Electrochemical /osmotic liquid waste nutrient extraction techniques should be reviewed more thoroughly in another paper.

Author Response

Open Review

English language and style

( ) Extensive editing of English language and style required 
(x) Moderate English changes required 
( ) English language and style are fine/minor spell check required 
( ) I don't feel qualified to judge about the English language and style 

Is the work a significant contribution to the field?

Is the work well organized and comprehensively described?

Is the work scientifically sound and not misleading?

Are there appropriate and adequate references to related and previous work?

Is the English used correct and readable?

Comments and Suggestions for Authors

Sustainability-558992 Review

Comment# 1:

Title:

Given the lengthy discussion and numerous details and citations of vermicomposting (~ 6/17 pages), the authors should reconsider changing the title to include  ”… review of vermicomposting…”

Reply to Comments:

We revised the title as suggested but not added vermicomposting as it is one part of this review. Since it is more economic friendly and sustainable, so we presented this in more detail.

Comment# 2:

Abstract

Long, with several misstatements or unclear sentences such as in lines 23-25”.  Nitrogen and potassium and either renewable or inexhaustible nutrients; whereas, Phosphorous is  non-renewable according to the  latest estimates of mine deposits.

The concluding sentence makes no sense, lines 30-33.

Comment# 3:

Body of the Manuscript

In general, very long, specially the introduction of nearly 3 single-spaced pages with general knowledge information about nutrients such as their well-known chemical forms and physiological functions.

Reply to Comments:

It has been revised all in the lights of suggestions.

Comment# 4:

Introduction

Lines 114-115.  Nitrate leaching does not cause soil acidification, but the conversion of ammonia to nitrate does. Excessive leaching can cause soil acidification over time with the loss of base cations, clay minerals….

Reply to Comment:  

This whole paragraph has been deleted.

Comment# 5:

NRP section

Composting a very common method of green and manure waste recycling, yet the authors do not give it sufficient consideration in light of the amount of space and information that they give to vermicomposting.

In other words,  what are the major advantages and disadvantages of both methods: composting and vermicomposting.

Again, the authors dedicate many pages to describing in detail vermicomposting but only about ½ page on composting with a handful of citations compared to ~20 citations on vermicomposting.  Therefore, the reader is not exposed to a comparative review of these two nutrient recovery processes.

Reply to Comment:  

Since vermicomposting is more climate friendly and sustainable so it was given more weightage. However, in the light of comments it has been revised.

Comment# 6:

NRT section

Confusing with information on biochar production together with plant nutrient productivity and GH emissions formulas and calculations (not sure what is the aim of this section)

Reply to Comment:  

NRT section describes pathway for nutrient recovery from wastes and includes three steps i.e. nutrient accumulation, release and extraction. Nutrient recovery technologies such as Pyrolysis, Forward osmosis and Electrodialysis have been described in this section.

Comment# 7:

There is also a table comparing management strategies and nutrient uptakes.  But it is far from complete since only 8 citations are in the table that covers composting, vermicomposting and biochar and only cover a very limited number of wastes.

Reply to Comment:  

Agreed but whatever relevant was available we added.

Comment# 8:

The sections on forward osmosis and electrodialysis seemed misplaced and are more appropriate to an environmental engineering paper.

Reply to Comment:  

We disagree as it is relevant to this also.

Comment# 9:

These two techniques may be innovative (based on old electrochemical principles) but far from being practical when applied to nutrient recoveries from solid mixed organic wastes and have yet to progress beyond laboratory studies, unlike composting and more recently biochar production techniques. These techniques limited to liquids and thus best suited for concentrated industrial waste streams and far from being optimal for complex mostly organic municipal wastewaters. 

Reply to Comment:  

Agreed that these techniques are far from being practical when applied to nutrient recoveries from solid mixed organic wastes, but it gives information about recovery of nutrients from other wastes.

Comment# 10:

LCA section and conclusions

Not sure what this section brings to the manuscript and there are not clear distinctions made since all processes, have carbon emissions footprint.  The authors fail to compare composting techniques with biochar. 

Which processes have the lowest initial, mid and long-term carbon foot-print?

Which techniques limit carbon emissions best and which capture carbon multigene rationally? And finally, which techniques and most cost-effective? (which drives their large-scale implementation beyond lab and field studies).

Reply to Comment:  

We appreciate reviewer comments. However, since comparison was not task of this review so we are not going to recommend any techniques based upon carbon emissions as here we are just presenting which methods/techniques could be used. This can be considered in our future articles as separate manuscript.

Comment# 11:

Conclusions, no section identified

Reply to Comment:  

Conclusion has been added now.

Comment# 12:

Lines 481 to 484 conclude the manuscript with two vague statements about waste management, threats to the environment and economic analysis.  

Nothing is stated about which of the “innovative” processes mentioned in the manuscript is(are) “best or promising or cost-effective….” to limit environmental damage, nutrient recovery, carbon footprint….etc.….

Reply to Comment:  

This has been deleted.

Comment# 13:

Recommendations

Authors should focus on solid waste methods of nutrient recycling… primarily composting, vermicomposting, municipal sludges, and biochar. Suggestion: add biosolids too, since these are produced in significant amounts by municipalities and an excellent source of N,P, and micronutrients.

They should endeavor to provide a detailed comparison of these systems of nutrient recovery which should include:

Applicability to types of wastes, complexity, recovery/loss rates (of nutrients), carbon footprint (short and long-terms benefits to carbon storage, life cycle impact, costs, scalability.

Electrochemical /osmotic liquid waste nutrient extraction techniques should be reviewed more thoroughly in another paper.

Reply to Comment:  

Agreed with reviewer and thanks for suggestions we will consider these points in another paper.

Round 2

Reviewer 1 Report

The authors have addressed all my comments. 

Author Response

Comments and Suggestions for Authors

The authors have addressed all my comments. 

Reply to Comment:

Thanks to the reviewer for positive remarks.

Reviewer 2 Report

This revised version offers few changes/additions that answer this reviewer's original concerns and comments.

Major emphasis on vermicomposting remains while other established methods are lightly reviewed and not explicitly compared to vermicomposting.

Emerging technologies are presented is such as way that they may be construed as mainstream (like composting) though they remain bench scale experimental technologies.

Thus the manuscript is essentially unchanged and the previous recommendation stands.

See files with some sentences highlighted as confusing and/or with typos.

Author Response

Open Review
(x) I would not like to sign my review report
( ) I would like to sign my review report
English language and style
( ) Extensive editing of English language and style required
(x) Moderate English changes required
( ) English language and style are fine/minor spell check required
( ) I don't feel qualified to judge about the English language and style
Is the work a significant contribution to the field?
Is the work well organized and comprehensively described?
Is the work scientifically sound and not misleading?
Are there appropriate and adequate references to related and previous work?
Is the English used correct and readable?
Comments and Suggestions for Authors
This revised version offers few changes/additions that answer this reviewer's original concerns and comments. Major emphasis on vermicomposting remains while other established methods are lightly reviewed and not explicitly compared to vermicomposting. Emerging technologies are presented is such as way that they may be construed as mainstream (like composting) though they remain bench scale experimental technologies. Thus, the manuscript is essentially unchanged, and the previous recommendation stands. See files with some sentences highlighted as confusing and/or with typos.
Reply to Comment:
Details of vermicomposting deleted as suggested. Please see Line#193 to 208. Similarly, all corrections incorporated as suggested. Detail about composting with comparison between composting and vermicomposting have been added now.
REPLY TO THE COMMENTS IN THE MAIN TEXT
Abstract
Comment#1
your mean "well established and new/emerging technologies."
Reply to comment:
Added as suggested. Kindly see the Line#21.
Comment#2
LINE#24
?? meaning of this sentence here. We know that plants extract/accumulate nutrients.
Reply to comment:
This line has been deleted with addition of modified statement.
Comment#3
LINE#30
Again, this are new emerging technologies bench scale lab. and should not be lumped with other proven technologies such as composting and biochar production.
Reply to comment:
Line has been rewritten as suggested
“Thus well proven technologies such as biochar, composting, vermicomposting, composting with biochar, pyrolysis and new emerging technologies (forward osmosis, and electro-dialysis) have potential to recover nutrients from wastes”.
Comment#4
LINE#33-36
Poor grammar, confusing sentence. Rewrite.
Reply to comment:
This has been deleted and rewritten new. Please see this
“Since waste management is big concern all over the globe and technologies e.g. landfill, combustion, incineration, pyrolysis, gasification are available to manage generated wastes but they have adverse impacts on society and on the environment. Thus, climate friendly technologies such as composting, biodegradation and anaerobic decomposition with generation of nonbiodegradable wastes need to be opted to have sustainable future environment. Furthermore, environmental impacts of technology could be quantified by life cycle assessment (LCA). Therfore, LCA could be used to evaluates performance of different environment-friendly technologies in waste managements and in the designing of future policies.”

Comment#5
Reply to comment:
LINE#321
so how good is vermicomposting in killing pathogens? Not clear which method is best from this discussion.
In spite of the extensive review of the literature on vermicomposting provided in this paper, this last paragraph does not clarify whether vermicomposting is superior to composting to kill pathogens

With Reference to old reviewer concerns (Round#1) more detail about composting have been added as well as comparisons between composting vs vermicomposting have been added now. Kindly see following paragraphs at Line#153 to 167.
Bioconversion of organic matter into humus like material called compost is called composting. This process occurs naturally if required conditions are available. Compost can be used as soil conditioner as discarded wastes have been decomposed to be act as a source of nutrients. Similarly, it can reduce (18%) amount of wastes entering to landfills. Good source of compost could be urban wastes which consist of vegetable matter and dead animal. Roughly urban areas can generate wastes in the range of 400-800 g per person per day and if city have population of five million it could generate wastes greater than 2500 tons per day. Thus, composting could be used as major portion of this wastes would be vegetables and putrescible matter. Composting can convert major portion of solid wastes into a marketable product. Compost generated through this process could help to improve the texture of light sandy soil, increased water retention, enlarging root system, availability (NPK typical percentage in compost; N, 1.2%; P, 0.7%; K, 1.2%) and uptake of nutrients. Composting is one of the best known recycling processes for the formation of soil conditioner from organic wastes. It is good way of cradle-to-cradle approach as it gives back organic matter to soil earlier taken by plants. This could help to have continue production of healthy crops for sustainable development.

Similarly, comparison between composting vs vermicomposting. See following revised paragraph (Line#353-359)
As vermicomposting method includes earthworms so its operating temperature ranges between 30-35oC, temperature above this level may prove lethal to earthworms that may stop the bioconversion process [37] thus less effective to kill the pathogens. A comparison of end product quality between composting vs. Vermicomposting were studied by using municipal compost (MC), municipal vermicompost (MV) and backyard vermicompost (BV) (Table 3-4). Microbial biomass-C, Dehydrogenase activity was higher in MC while hydrolase activities (urease, protease and phosphatase) were higher in the vermicomposts than in the municipal compost (Table 4). The result further showed that vermicompost had significantly larger nutrient concentrations than the compost when mixed with the soil as well as higher microbial populations size and activity and increased crop yield [60].
Table 3 Composting vs Vermicomposting.
Composting Vermicomposting
Depth Can be any depth Worms usually prefer to live in the top 6″ – 12″ of the bedding (cannot be deep)
Convenience Outdoors only with specialized buildings and equipment outdoors or indoors
Speed Hot composting takes 6 – 9 months to produce fertilizer Much faster
Heat Levels Hot as the aerobic breakdown of organic matter releases carbon dioxide and heat, resulting in piles than can top 70°C Cooler process with temperatures ranging between 10-32°C
Microbial Populations Dominated by thermophilic (or “heat-loving”) microbes Dominated by mesophilic microbes
Aeration Turning is required Turning is not required
Cost Cheap Needs care for worms protections
Financial Value Cheap Much greater financial value

Table 4 Biological and biochemical properties of composts and vermicompost [60].
Municipal compost (MC) Municipal vermicompost (MV) Backyard vermicompost (BV)
Microbial biomass C (μg C g-1) 1147a 703b 335c
Urease (μmol NH4+ g-1h-1) 3.54b 3.9b 6.11a
BAA-Protease (μmol NH4+ g-1h-1) 0.31c 0.96b 1.83a
Phosphatse (μmol PNP g-1h-1) 237c 398b 676a
Dehyrogenase (μg INTF g-1) 193a 123b 77c

Comment#6
LINE#378
Not clear why this is needed here.
Reply to comment:
This has been deleted. Kindly see LINE#413 to 429.
Comment#7
LINE#458
Poor sentence (tim=typo?)
Reply to comment:
Corrected
Comment#8
LINE#468
Poor/confusing sentences. Rewrite.

Reply to comment:
Corrected as suggested. Kindly see LINE#507 to 510.
